# TD LEARNING WITH CONSTRAINED GRADIENTS

## ABSTRACT

Temporal Difference Learning with function approximation is known to be unstable. Previous work like Sutton et al. (2009b) and Sutton et al. (2009a) has presented alternative objectives that are stable to minimize. However, in practice, TD-learning with neural networks requires various tricks like using a target network that updates slowly (Mnih et al., 2015). In this work we propose a constraint on the TD update that minimizes change to the target values. This constraint can be applied to the gradients of any TD objective, and can be easily applied to non-linear function approximation. We validate this update by applying our technique to deep Q-learning, and training without a target network. We also show that adding this constraint on Baird's counterexample keeps Q-learning from diverging.

## 1 INTRODUCTION

Temporal Difference learning is one of the most important paradigms in Reinforcement Learning (Sutton & Barto). Techniques based on nonlinear function approximators and stochastic gradient descent such as deep networks have led to significant breakthroughs in the class of problems that these methods can be applied to (Mnih et al., 2013; 2015; Silver et al., 2016; Schulman et al., 2015). However, the most popular methods, such as TD($\lambda$), Q-learning and Sarsa, are not true gradient descent techniques (Barnard, 1993) and do not converge on some simple examples (Baird et al., 1995).

Baird et al. (1995) and Baird & Moore (1999) propose residual gradients as a way to overcome this issue. Residual methods, also called backwards bootstrapping, work by splitting the TD error over both the current state and the next state. These methods are substantially slower to converge, however, and Sutton et al. (2009b) show that the fixed point that they converge to is not the desired fixed point of TD-learning methods.

Sutton et al. (2009b) propose an alternative objective function formulated by projecting the TD target onto the basis of the linear function approximator, and prove convergence to the fixed point of this projected Bellman error is the ideal fixed point for TD methods. Bhatnagar et al. (2009) extend this technique to nonlinear function approximators by projecting instead on the tangent space of the function at that point. Subsequently, Scherrer (2010) has combined these techniques of residual gradient and projected Bellman error by proposing an oblique projection, and Liu et al. (2015) has shown that the projected Bellman objective is a saddle point formulation which allows a finite sample analysis.

However, when using deep networks for approximating the value function, simpler techniques like Q-learning and Sarsa are still used in practice with stabilizing techniques like a target network that is updated more slowly than the actual parameters (Mnih et al., 2015; 2013).

In this work, we propose a constraint on the update to the parameters that minimizes the change to target values, *freezing* the target that we are moving our current predictions towards. Subject to this constraint, the update minimizes the TD-error as much as possible. We show that this constraint can be easily added to existing techniques, and works with all the techniques mentioned above.

We validate our method by showing convergence on Baird's counterexample and a gridworld domain. On the gridworld domain we parametrize the value function using a multi-layer perceptron, and show that we do not need a target network.

## 2    NOTATION AND BACKGROUND

Reinforcement Learning problems are generally defined as a Markov Decision Process (MDP), $(\mathcal{S}, \mathcal{A}, P, \mathcal{R}, R, d_0, \gamma)$. We use the definition and notation as defined in Sutton & Barto, second edition, unless otherwise specified.

In case of a function approximation, we define the value and action value functions with parameters by $\theta$.

$$v_\pi(s|\theta) \doteq \mathbb{E}_\pi \left[ R_t + \gamma R_{t+1} + \gamma^2 R_{t+2} + \ldots | S_t = s \right] \tag{1}$$

$$q_\pi(s, a|\theta) \doteq \mathbb{E}_\pi \left[ R_t + \gamma R_{t+1} + \gamma^2 R_{t+2} + \ldots | S_t = s, A_t = a \right] \tag{2}$$

We focus on TD(0) methods, such as Sarsa, Expected Sarsa and Q-learning. The TD error that all these methods minimize is as follows:

$$\delta_{TD} = v_\pi(s_t|\theta) - (r_t + \gamma v_{\pi'}(s_{t+1}|\theta)) \tag{3}$$

The choice of $\pi'$ determines if the update is on-policy or off-policy. For Q-learning the target is $\max_a q(s_{t+1}, a)$.

If we consider TD-learning using function approximation, the loss that is minimized is the squared TD error. For example, in Q-learning

$$\mathcal{L}_{TD} = \|q(s_t, a_t|\theta) - r_t - \gamma \max_a q(s_{t+1}, a|\theta)\|^2$$

The gradient of this loss is the direction in which you update the parameters. We shall define the gradient of the TD loss with respect to state $s_t$ and parameters $\theta_t$ as $g_{TD}(s_t)$. The gradient of some other function $f(s_t|\theta_t)$ can similarly be defined as $g_f(s_t)$. The parameters are then updated according to gradient descent with step size $\alpha$ as follows:

$$g_{TD}(s_t) = \frac{\partial \mathcal{L}_{TD}}{\partial s_t} \frac{\partial s_t}{\partial \theta_t} \tag{4}$$

$$\theta_{t+1} = \theta_t - \alpha g_{TD}(s_t) \tag{5}$$

## 3    TD UPDATES WITH CONSTRAINED GRADIENTS

A key characteristic of TD-methods is bootstrapping, i.e. the update to the prediction at each step uses the prediction at the next step as part of it's target. This method is intuitive and works exceptionally well in a tabular setting (Sutton & Barto). In this setting, updates to the value of one state, or state-action pair do not affect the values of any other state or state-action.

TD-learning using a function approximator is not so straightforward, however. When using a function approximator, states nearby will tend to share features, or have features that are very similar. If we update the parameters associated with these features, we will update the value of not only the current state, but also states nearby that use those features. In general, this is what we want to happen. With prohibitively large state spaces, we want to generalize across states instead of learning values separately for each one. However, if the value of the state visited on the next step, which often does share features, is also updated, then the results of the update might not have the desired effect on the TD-error.

Generally, methods for TD-learning using function approximation do not take into account that updating $\theta_t$ in the direction that minimizes TD-error the most, might also change $v(s_{t+1}|\theta_{t+1})$.
Though they do not point out this insight as we have, previous works that aims to address convergence of TD methods using function approximation do deal with this issue indirectly, like residual gradients (Baird et al., 1995) and methods minimizing MSPBE (Sutton et al., 2009b). Residual gradients does this by essentially updating the parameters of the next state in the opposite direction of the update to the parameters of the current state. This splits the error between the current state and the next state, and the fixed point we reach does not act as a predictive representation of the reward. MSPBE methods act by removing the component of the error that is not in the span of the features of the current state, by projecting the TD targets onto these features. The update for these methods

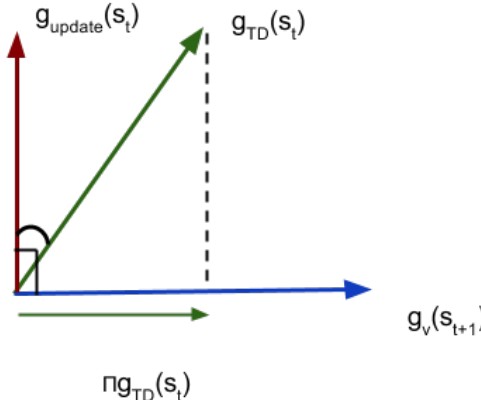

Figure 1: Modifying the gradient by projecting onto the direction orthogonal to direction of gradient at $s_{t+1}$

involves the product of three expectations, which is handled by keeping a separate set of weights that approximate two of these expectations, and is updated at a faster scale. The idea also does not immediately scale to nonlinear function approximation. Bhatnagar et al. (2009) propose a solution by projecting the error on the tangent plane to the function at the point at which it is evaluated.

### 3.1 CONSTRAINING THE UPDATE

We propose to instead constrain the update to the parameters such that the change to the values of the next state is minimized, while also minimizing the TD-error. To do this, instead of modifying the objective, we look at the gradients of the update.

$g_{TD}(s_t)$ is the gradient at $s_t$ that minimizes the TD error. $g_v(s_{t+1})$ is the gradient at $s_{t+1}$ that will change the value the most. We update the parameters $\theta_t$ with $g_{update}(s_t)$ such that the update is orthogonal to $g_v(s_{t+1})$. That is, we update the parameters $\theta_t$ such that there is no change in the direction that will affect $v(s_{t+1})$. Graphically, the update can be seen in figure 1. The actual updates to the parameters are as given below.

$$g_{update}(s_t) = g_{TD}(s_t) - \Pi g_{TD}(s_t) \tag{6}$$

$$\hat{g}_v(s_{t+1}) = \frac{g_v(s_{t+1})}{\|g_{TD}(s_{t+1})\|} \tag{7}$$

$$\Pi g_{TD}(s_t) = (g_{TD}(s_t) \cdot \hat{g}_v(s_{t+1})) \times \hat{g}_v(s_{t+1}) \tag{8}$$

$$\theta_{t+1} = \theta_t - \alpha g_{update}(s_t) \tag{9}$$

As can be seen, the proposed update is orthogonal to the direction of the gradient at the next state. Which means that it will minimize the impact on the next state. On the other hand, $\angle(g_{update}(s_t), g_{TD}(s_t)) \leq 90°$. This implies that applying $g_{update}(s_t)$ to the parameters $\theta$ minimizes the TD error, unless it would change the values of the next state.

Furthermore, our technique can be applied on top of any of these techniques to improve their behavior. We show this for residual gradients and Q-learning in the following experiments.

## 4 EXPERIMENTS

To show that our method learns just as fast as TD while guaranteeing convergence similar to residual methods, we show the behavior of our algorithm on the following 3 examples.

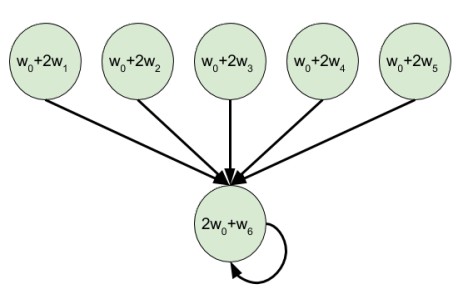
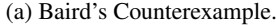

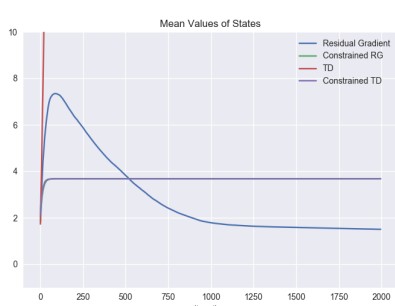

(a) Baird's Counterexample.

(b) Comparison of the average values across states on Baird's counterexample over first 2000 iterations of training

Figure 2: Baird's Counterexample is specified by 6 states and 7 parameters. The value of each state is calculated as given inside the state. At each step, the agent is initialized at one of the 6 states uniformly at random, and transitions to the state at the bottom, shown by the arrows.

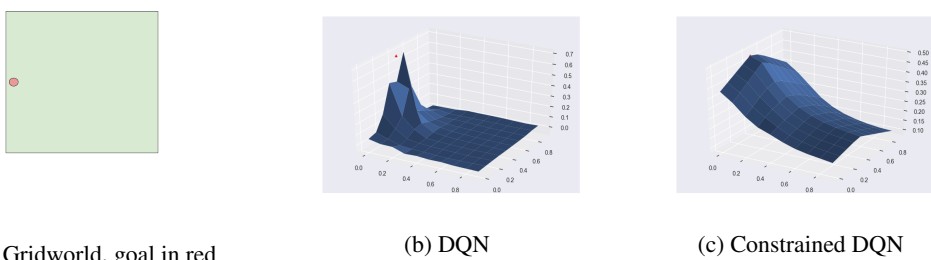

(a) Gridworld, goal in red

(b) DQN

(c) Constrained DQN

Figure 3: A $10 \times 10$ Gridworld with a goal at location $(0, 4)$, which is midway between one of the walls. Both DQN and Constrained DQN are used to approximate the value function for a softmax policy.

## 4.1 BAIRD'S COUNTEREXAMPLE

Baird's counterexample is a problem introduced in Baird et al. (1995) to show that gradient descent with function approximation using TD updates does not converge.

The comparison of our technique with Q-learning and Residual Gradients is shown in figure 2. We compare the average performance for all tehcniques over 10 independent runs.

If we apply gradient projection while using the TD error, we show that both Q-learning (TD update) and updates using residual gradients (Baird et al., 1995) converge, but not to the ideal values of 0. In the figure, these values are almost overlapping. Our method constrains the gradient to not modify the weights of the next state, which in this case means that $w_0$ and $w_6$ never get updated. This means that the values do not converge to the true values (0), but they do not blow up as they do if using regular TD updates. Residual gradients converge to the ideal values of 0 eventually. GTD2 (Sutton et al., 2009b) also converges to 0, as was shown in the paper, but we have not included that in this graph to avoid cluttering.

## 4.2 GRIDWORLD

The Gridworld domain we use is a $(10 \times 10)$ room with $d_0 = \mathcal{S}$, and $R((0, 4)) = 1$ and 0 everywhere else. We have set the goal as $(0, 4)$ arbitrarily and our results are similar for any goal on this grid.

The input to the function approximation is only the $(x, y)$ coordinates of the agent. We use a deep network with 2 hidden layers, each with 32 units, for approximating the Q-values. We execute a softmax policy, and the target values are also calculated as $v(s_{t+1}) = \sum_a \pi(a|s_{t+1})q(s_{t+1}, a)$,

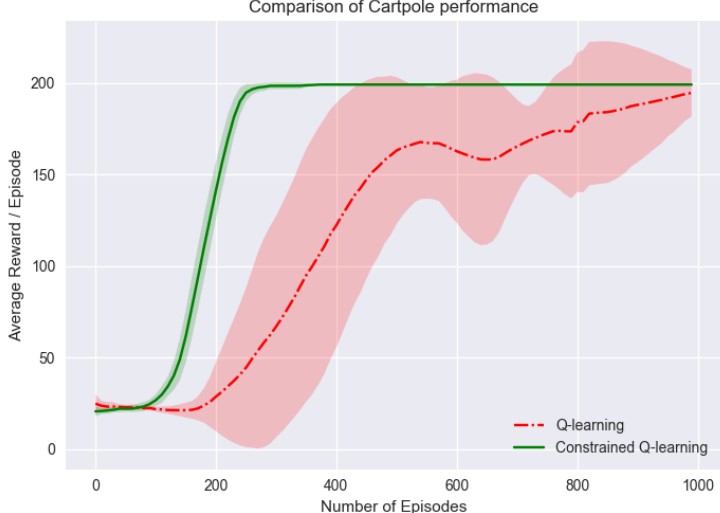

Figure 4: Comparison of DQN and Constrained on the Cartpole Problem, taken over 10 runs. The shaded area specifies std deviation in the scores of the agent across independent runs. The agent is cut off after it's average performance exceeds 199 over a running window of 100 episodes

where the policy $\pi$ is a softmax over the Q-values. The room can be seen in Figure 3 with the goal in red, along with a comparison of the value functions learnt for the 2 methods we compare.

| - | Q-learning | Constrained Q-learning |
|---|---|---|
| MSE | $0.0335 \pm 0.0017$ | $\mathbf{0.0076 \pm 0.0028}$ |

Table 1: Comparison of the Mean Squared Error between the value function approximated by Q learning and by Constrained Q learning with respect to the value function calculated by policy evaluation on the Gridworld domain. Constrained Q-learning gets substantially lower error.

We see from the learned value function that the value function that DQN learns is sharper. This might be because the next state values that it uses to update are from a target network that updates slowly and thus provides stale targets. Constraining the update leads to a smoother value function, which is encouraging since it shows that constraint does not dissuade generalization. This experiment shows that constrained updates allow generalization that is useful, while not allowing the target to drift off or values to explode.

The ground truth can be calculated for this domain using tabular policy evaluation. We calculate this ground truth value function and compare Mean Squared Error with the value functions learned by DQN and Constrained DQN over 10 independent runs. The results of this comparison can be seen in table 1

## 4.3 CARTPOLE

As a way to compare against Q-learning with a deep network, we test on the cartpole domain (Barto et al., 1983). We use implementations from OpenAI baselines (Hesse et al., 2017) for Deep Q-learning to ensure that the code is reproducible and to ensure fairness. The network we use is with 2 hidden layers of $[5, 32]$. The only other difference compared to the implemented baseline is that we use RMSProp (Tieleman & Hinton, 2012) as the particular machinary for optimization instead of Adam (Kingma & Ba, 2014). This is just to stay close to the method used in Mnih et al. (2015) and in practice, Adam works just as well and the comparison is similar.

The two methods are trained using exactly the same code except for the updates, and the fact that Constrained DQN does not use a target network. We can also train COnstrained DQN with a larger

step size ($10^{-3}$), while DQN requires a smaller step size ($10^{-4}$) to learn. The comparison with DQN is shown in 4. We see that constrained DQN learns much faster, with much less variance than regular DQN.

## 5 DISCUSSION AND CONCLUSION

In this paper we introduce a constraint on the updates to the parameters for TD learning with function approximation. This constraint forces the targets in the Bellman equation to not move when the update is applied to the parameters. We enforce this constraint by projecting the gradient of the TD error with respect to the parameters for state $s_t$ onto the orthogonal space to the gradient with respect to the parameters for state $s_{t+1}$.

We show in our experiments that this added constraint stops parameters in Baird's counterexample from exploding when we use TD-learning. But since we do not allow changes to target parameters, this also keeps Residual Gradients from converging to the true values of the Markov Process.

On a Gridworld domain we demonstrate that we can perform TD-learning using a 2-layer neural network, without the need for a target network that updates more slowly. We compare the solution obtained with DQN and show that it is closer to the solution obtained by tabular policy evaluation. Finally, we also show that constrained DQN can learn faster and with less variance on the classical Cartpole domain.

For future work, we hope to scale this approach to larger problems such as the Atari domain (Bellemare et al., 2013). We would also like to prove convergence of TD-learning with this added constraint.

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
