# OpenReview forum: "TD Learning with Constrained Gradients"
_ICLR.cc/2018/Conference — Reject_

### Official Review · AnonReviewer1 · 2017-11-27
**A new approach to off-policy TD with function approximation**

**Rating:** 2
**Confidence:** 4

**Review:**

Summary: This paper tackles the issue of combining TD learning methods with function approximation. The proposed algorithm constrains the gradient update to deal with the fact that canonical TD with function approximation ignores the impact of changing the weights on the target of the TD learning rule. Results with linear and non-linear function approximation highlight the attributes of the method.

Quality: The quality of the writing, notation, motivation, and results analysis is low. I will give a few examples to highlight the point. The paper motivates that TD is divergent with function approximation, and then goes on to discuss MSPBE methods that have strong convergence results, without addressing why a new approach is needed. There are many missing references: ETD, HTD, mirror-prox methods, retrace, ABQ. Q-sigma. This is a very active area of research and the paper needs to justify their approach. The paper has straightforward technical errors and naive statements: e.g. the equation for the loss of TD takes the norm of a scalar. The paper claims that it is not well-known that TD with function approximation ignores part of the gradient of the MSVE. There are many others.

The experiments have serious issues. Exp1 seems to indicate that the new method does not converge to the correct solution. The grid world experiment is not conclusive as important details like the number of episodes and how parameters were chosen was not discussed. Again exp3 provides little information about the experimental setup.

Clarity: The clarity of the text is fine, though errors make things difficult sometimes. For example The Bhatnagar 2009 reference should be Maei.

Originality: As mentioned above this is a very active research area, and the paper makes little effort to explain why the multitude of existing algorithms are not suitable.

Significance: Because of all the things outlined above, the significance is below the bar for this round.

---

### Official Review · AnonReviewer2 · 2017-11-27
**Issues with justification for constrained update**

**Rating:** 3
**Confidence:** 4

**Review:**

This paper proposes adding a constraint to the temporal difference update to minimize the effect of the update on the next state’s value. The constraint is added by projecting the original gradient to the orthogonal of the maximal direction of change of the next state’s value. It is shown empirically that the constrained update does not diverge on Baird’s counter example and improves performance in a grid world domain and cart pole over DQN.

This paper is reasonably readable. The derivation for the constraint is easy to understand and seems to be an interesting line of inquiry that might show potential.

The key issue is that the justification for the constrained gradients is lacking. What is the effect, in terms of convergence, in modifying the gradient in this way? It seems highly problematic to simply remove a whole part of the gradient, to reduce effect on the next state. For example, if we are minimizing the changes our update will make to the value of the next state, what would happen if the next state is equivalent to the current state (or equivalent in our feature space)? In general, when we project our update to be orthogonal to the maximal change of the next states value, how do we know it is a valid direction in which to update?

I would have liked some analysis of the convergence results for TD learning with this constraint, or some better intuition in how this effects learning. At the very least a mention of how the convergence proof would follow other common proofs in RL. This is particularly important, since GTD provides convergent TD updates under nonlinear function approximation; the role for a heuristic constrained TD algorithm given convergent alternatives is not clear.

For the experiments, other baselines should be included, particularly just regular Q-learning. The primary motivation comes from the use of a separate target network in DQN, which seems to be needed in Atari (though I am not aware of any clear result that demonstrates why, rather just from informal discussions). Since you are not running experiments on Atari here, it is invalid to simply assume that such a second network is needed. A baseline of regular Q-learning should be included for these simpler domains.

The results in Baird’s counter example are discouraging for the new constraints. Because we already have algorithms which better solve this domain, why is your method advantageous? The point of showing your algorithm not solve Baird’s counter example is unclear.

There are also quite a few correctness errors in the paper, and the polish of the plots and language needs work, as outlined below.

There are several mistakes in the notation and background section.
1. “If we consider TD-learning using function approximation, the loss that is minimized is the squared TD error.“ This is not true; rather, TD minimizes the mean-squared project Bellman error. Further, L_TD is strangely defined: why a squared norm, for a scalar value?
2. The definition of v and delta_TD w.r.t. to v seems unnecessary, since you only use Q. As an additional (somewhat unimportant) point, the TD-error is usually defined as the negative of what you have.
3. In the function approximation case the value function and q functions parameterized by \theta are only approximations of the expected return.
4. Defining the loss w.r.t. the state, and taking the derivative of the state w.r.t. to theta is a bit odd. Likely what you meant is the q function, at state s_t? Also, are ignoring the gradient of the value at the next step? If so, this further means that this is not a true gradient.

There is a lot of white space around the plots, which could be used for larger more clear figures. The lack of labels on the plots makes them hard to understand at a glance, and the overlapping lines make finding certain algorithm’s performance much more difficult. I would recommend combining the plots into one figure with a drawing program so you have more control over the size and position of the plots.

Examples of odd language choices:
	-	“The idea also does not immediately scale to nonlinear function approximation. Bhatnagar et al. (2009) propose a solution by projecting the error on the tangent plane to the function at the point at which it is evaluated. “ - The paper you give exactly solves for the nonlinear function approximation case. What do you mean does not scale to nonlinear function approximation? Also Maei is the first author on this paper.
	-	“Though they do not point out this insight as we have” - This seems to be a bit overreaching.
- “the gradient at s_{t+1} that will change the value the most”  - This is too colloquial. I think you simply mean the gradient of the value function, for the given s_t, but its not clear.

---

### Official Review · AnonReviewer3 · 2017-11-27
**Interesting but not enough supporting the idea**

**Rating:** 4
**Confidence:** 4

**Review:**

This is an interesting idea, and written clearly. The experiments with Baird's and CartPole were both convincing as preliminary evidence that this could be effective. However, it is very hard to generalize from these toy problems. First, we really need a more thorough analysis of what this does to the learning dynamics itself. Baring theoretical results, you could analyze the changes to the value function at the current and next state with and without the constraint to illustrate the effects more directly. I think ideally, I would want to see this on Atari or some of the continuous control domains often used. If this allows the removing of the target network for instance, in those more difficult tasks, then this would be a huge deal.

Additionally, I do not think the current gridworld task adds anything to the experiments, I would rather actually see this on a more interesting linear function approximation on some other simple task like Mountain Car than a neural network on gridworld. The reason this might be interesting is that when the parameter space is lower dimensional (not an issue for neural nets, but could be problematic for linear FA) the constraint might be too much leading to significantly poorer performance. I suspect this is the actual cause for it not converging to zero for Baird's, although please correct me if I'm wrong on that.

As is, I cannot recommend acceptance given the current experiments and lack of theoretical results. But I do think this is a very interesting direction and hope to see more thorough experiments or analysis to support it.

Pros:
Simple, interesting idea
Works well on toy problems, and able to prevent divergence in Baird's counter-example

Cons:
Lacking in theoretical analysis or significant experimental results

---

### Public Comment · (anonymous) · 2017-11-28
**Availability of codes**

Hi I am trying to repeoduce your results. Is it possible to share the codes for this work?

---

> ### Public Comment · (anonymous) · 2017-12-03
> **Don't waste your time**
>
> The algorithm does not generalize to anything more complicated than toy environments like Cartpole, as multiple reviewers have pointed out. I'm happy to be proven wrong, but I strongly doubt it would help your real task.

---

### Public Comment · (anonymous) · 2017-12-14
**summary of a reproducibility  study on this paper**

In efforts to ensure that published results are reliable and reproducible in Machine Learning research, we investigated the reproducibility of empirical results of this paper. We tried to reproduce the experimental results shown in the paper. However, there were some difficulties we faced.

(1) The notation and equations shown in the paper lack of clarity. For example, the authors did not mathematically define the variable  g_v(s_{t+1}), they described it as the gradient at s_{t+1} that will change the value most. After some research, we found out that the authors have replaced the g_v(s_{t+1}) with gTD(s_{t+1}) in their revised version of this paper that was submitted to the Deep Reinforcement Learning Symposium at NIPS. Only after this discovery were we able to proceed with the implementation.
(2) There are no clear mention of how they are calculating g_{TD}(s_{t+1}) which seems to be the gradient of the TD error with respect to the next state. However, after communicating with the authors we found that  g_{TD}(s_{t+1}) is the gradient of the same TD error with respect to the target and all the experiments are done by taking a single step within the environment. We had difficulties in figuring out how this translates to Q learning because the target is a max operator applied to the next state action pair. Under such circumstances we will not be able to differentiate with respect to the target.
(3) There were no code available for the experiments by the time we finished this report. We discussed with the authors regarding the availability of the code base and was assured that it is going to be released soon.
(4) The authors did not report all the hyper-parameters they used in their experiments.

Although we were not able to fully duplicate the experiments  due to the above reasons, we would like to share what we did and our findings. You can check out the full report at https://www.overleaf.com/read/tdzyfmjzkhyj
(1) Cartpole : we used the exact set of hyper-parameters reported by the authors. In addition, we used the default open-AI batch size as it is not mentioned by the authors. The baseline we got is quite different than the one of the authors'. However, interestingly, a model with single hidden layer of 64 units got us a baseline result that is as good as the results claimed by the authors.
(2) GridWorld : We have run DQN in the 10x10 Grid World environment as proposed by the authors. Since the authors did not mention the starting point they used, we set it to be (0,0). For DQN we used  two hidden layers, units of size 32 per hidden layer. We executed a soft max policy and feed the (x,y) coordinates of the agent in the network. As the authors did not mention the total episodes they ran. we therefore ran them over 1000 episodes and took an average over 10 independent runs as before. We computed the Q values for DQN with that of the value function obtained by running policy evaluation in this domain, and obtained a mean squared error around 0.38. Note that we only verified the DQN baseline, we did not verify the proposed algorithm in the DQN setup.
(3) Baird's counterexample : We ran both TD and constrained TD on the Baird-6-state setup for 2000 steps each run and we made 10 independent runs.  We set the discount factor to be 0.99, the learning rate to be 0.01. In addition we extracted the feature values from the graph shown in the paper. We initialized the weights the same way Sutton did in his book for the Baird's counterexample section.
We observed the diverging behavior when running regular TD. We obtained a similar baseline to that mentioned in the paper.  Nevertheless, we found that our constrained TD produced a quite different shape, a bell shape, heavy tail curve compared to  a converging straight line after iteration 100 reported by the authors.
(4) linear function approximation : The authors claimed that the constraint can be applied to the gradients of any TD objective. Thus we also tried some experiments with linear function approximation in open-AI mountain car environment (max size of step =200 ). We made 10 independent runs and took the average. We ran both Q learning and SARSA and tried to implement the Constrained SARSA and Constrained Q learning.  In terms of Q learning the target is simply a max function over the next state action pair, and in such cases this max function is not differentiable. Further more we tried  SARSA as the target is differentiable. For our implementation we use RBF kernel for approximating value functions. We have incorporated a learning rate of 0.01 and a discount factor of 0.99 for our implementation. It is important to note that we observed no signs of learning for Constrained SARSA in this environment.
In conclusion, we hope that our above findings are helpful to the authors as well as people who are interested in this paper. We encourage the authors to publish their code and provide more details about hyper-parameters for their work.

---

> ### Author Response · Authors · 2018-01-05
> **Thank You for this Study**
>
> I would like to thank the commenter for their in-depth study of our algorithm.
> It is definitely very helpful in analyzing our approach and to guide our further inquiries.
> We apologize for not being very clear in our methodology or hyper-parameters.
> We will definitely make a more concerted effort to maintain reproducibility and also make sure to report all training conditions and hyper-parameters in the future.

---

### Decision · Program_Chairs · 2018-01-29
**ICLR 2018 Conference Acceptance Decision**

**Decision:**

Reject

**Comment:**

The reviewers agree this paper is not yet ready for publication.